# Corrosion Behavior of Reinforcing Steel in the Immersed Tube Tunnel (ITT) under Submarine Environment

**DOI:** 10.3390/ma16093300

**Published:** 2023-04-22

**Authors:** Yu Yan, Haiwei Zhu, Zhihong Fan, Jiaqi Zhao, Shuping Jiang

**Affiliations:** 1School of Civil Engineering, Chongqing Jiaotong University, Chongqing 400074, China; 2Key Laboratory of Harbor and Marine Structure Durability Technology of the Ministry of Communications, CCCC Fourth Harbor Engineering Institute Co., Ltd., Guangzhou 510230, China; fzhihong@cccc4.com (Z.F.); zjq626@outlook.com (J.Z.)

**Keywords:** submarine environment, immersed tube tunnel, reinforced concrete structure, corrosion behavior, reinforcing steel, electrochemical testing

## Abstract

The corrosion behavior of reinforcing steel in the ITT under a submarine environment was investigated. Electrochemical tests were carried out to separately determine the linear polarization curves and the AC impedance spectra of rebars in the ITT scaled-down models subjected to pressurized seawater erosion, from which key parameters were obtained, including the self-corrosion potential (*E_corr_*), corrosion current density (*i_corr_*), polarization resistance (*R_p_*), concrete resistance (*R_c_*), and charge migration resistance (*R_ct_*). The results show that in the process of pressurized seawater erosion, the rebars on the seawater side of the ITT models corroded earlier than the rebars on the cavity side, and it is recommended that anti-chloride ion penetration measures be taken on the surface of the seawater side as a priority in the project. The corrosion rate of rebars on the seawater side was significantly higher than that on the cavity side, and the corrosion rate of rebars on the cavity side increased as the erosion time increased. The corrosion rate of rebars in the ITT models was affected by chloride ions to a greater extent than by oxygen. Furthermore, by regression equation, a linear function between *R_p_* obtained from the polarization curves and *R_ct_* obtained from the AC impedance spectra was established.

## 1. Introduction

Since the first use of an ITT in the United States in 1910 in the Yerba Buena Island Crossing, more than 150 submarine tunnel projects in the world have adopted ITTs [1]. And in recent years, with the implementation of China’s national strategy of “Ocean Power”, long-service-life concrete projects, such as the Hong Kong–Zhuhai–Macao Bridge [2], the Shenzhen–Zhongshan Link [3], the Bohai Bay Submarine Tunnel [4], and the Jiaozhou Bay Submarine Tunnel [5] have been completed or are under planning. In addition, there is also the under–construction Fehmarnbelt Tunnel between Germany and Denmark, and these submarine tunnels all use ITTs as a construction option. Driven by the convenience of transportation and the development of sea resources, and secured by high–performance concrete preparation technologies, advanced engineering machines, and construction technologies, the ITT has the advantages of short construction time and high construction quality [6], and has been adopted by more and more submarine tunnel projects.

While under the submarine environment, the durability of ITTs, like other reinforced concrete (RC) structures, is mainly faced with the corrosion of reinforcing steel caused by seawater erosion, which will lead to structural performance degradation and eventual failure [7,8]. Therefore, the builders have put forward high requirements for the durability design of the ITTs, taking into account economics, safety, and long service life. For the corrosion mechanism of RC structures, researchers have found that environmental temperature [7], protective cover [9], and the pH of the concrete pore solution [10] have significant effects on the corrosion of reinforcing steel; in addition, it is also found that the corrosion process requires oxygen and that the thickness of the concrete protective layer does not affect oxygen penetration in a dry air environment [9]. In contrast, RC structures completely immersed in seawater, despite being in a high salt and high humidity environment, are less likely to corrode due to the difficulty of oxygen diffusion [9]. However, the ITTs differ from the two aforementioned cases under absolute humidity conditions. The various rebars in ITTs are usually electrically connected by stirrups, whereby macrocell corrosion effects are generated in the presence of aggressive media, such as chloride ions, oxygen, water, and carbon dioxide [11,12,13], resulting in extremely high corrosion rates [11] that deteriorate the tunnels. Regarding the macrocell corrosion in concrete structures under the marine environment, researchers have conducted studies by electrochemical testing techniques. For example, Chen [14] verified the applicability of the Stern–Geary equation for the macrocell corrosion of reinforced concrete, and Li [15] found that corrosion will occur earlier and damage the structure more severely at the site where the macrocell effect occurs, and Lliso−Ferrando [16] found that the poorer the concrete quality and the smaller the cover layer, the greater the effect of the macrocell. Therefore, it is important to investigate the electrochemical corrosion behavior of reinforcing steel in ITTs under submarine environments, which can provide suggestions for the durability design of ITTs.

Current research on ITTs under submarine environments is mainly concerned with the diffusive behavior of chloride ions [17,18,19], bearing capacity [20], seismic performance [21,22], and impact resistance performance [23], whereas relatively few studies have been conducted on using electrochemical methods to investigate the corrosion behavior of reinforcing steel in the tunnels. Unlike concrete structures buried deep in the sea that are fully saturated with seawater, there is typically air circulation in the cavities of ITTs, which makes it particularly important to consider the influence of the bidirectional diffusion of oxygen and chloride ions inside concrete on the corrosion of reinforcing steel. For the problem of reinforcing steel corrosion faced by ITTs in the marine environment, cathodic protection [24,25] is a common measure at present; although the protection effect is good, there are the disadvantages of complex installation, too fast consumption of anodes, and high maintenance costs. Epoxy-coated reinforcing steel [24,26] has good anti-corrosion performance, but there are also problems of high costs and broken epoxy coating leading to accelerated local corrosion. In addition, although the use of rust inhibitors [24,27,28], waterproof or hydrophobic materials [24,29], and Zn–Al coatings [24,30] can all effectively improve the durability of ITTs, problems of increasing construction costs or maintenance difficulties exist. Considering the importance of ITTs in transportation and the high cost of construction and maintenance, multiple anti-corrosion methods are usually used together to ensure the durability of ITTs, and no single method is used. Therefore, the significance of this study is to provide a more targeted approach to solve the corrosion problem of reinforcing steel in ITTs by revealing the corrosion law of reinforcing steel. In this study, electrochemical methods are used to separately determine the linear polarization curves and the AC impedance spectra of rebars in the ITT scaled-down models subjected to erosion by pressurized seawater, from which key parameters, including *E_corr_*, *i_corr_*, *R_p_*, *R_c_*, *R_ct_,* and the apparent diffusion coefficient (*n*), are obtained to systematically analyze the electrochemical corrosion behavior of rebars in the ITTs, so as to make suggestions for the durability design and maintenance of ITTs.

## 2. Materials and Methods

### 2.1. Raw Materials and Mix Ratio

To shorten the test time as well as to improve the reliability of the test data, mortars with large water-cement ratios were used. Mortars with large water-cement ratios have larger porosity, thus improving the transport of aggressive agents such as oxygen, water, and chloride ions [31], which can effectively shorten the test time. Although there are differences between the mortar used in this work and the concrete used in the actual project, both are close with respect to the transport behavior of aggressive agents [32,33] and the corrosion behavior of reinforcing steel [34,35].

This study was performed using P∙II 42.5 ordinary Portland cement (for which the chemical composition is shown in Table 1 and the physical and mechanical properties are shown in Table 2); ordinary river sand with an apparent density of 2639 kg/m^3^, a mud content of 0.7% (mass percentage) and a fineness modulus of 2.64; and tap water. The mix ratio and properties of the mortar are shown in Table 3.

### 2.2. Design and Fabrication of ITT Scaled-Down Models

Two ITT scaled-down models with lengths, widths, and heights of 300 mm, 300 mm, and 280 mm, respectively, were prepared and named as Model 1 and Model 2. A total of 16 pieces of 8-mm diameter HPB300 bright-round carbon rebars (for which the elemental composition is shown in Table 4) were buried in each model, where 8 pieces each were buried in the inner (near the cavity) and outer (near the seawater) sides, the thickness of the protective layer of the bars was set to 30 mm, and the wall thickness was 100 mm. The rebars to be tested were acid washed, soaked in saturated calcium hydroxide solution for 7 d, and then buried in the mortar together with saturated glycerol electrodes and Ag/AgCl probes. The ends of the rebars were soldered with copper conductors, and finally, the top and bottom of the tunnel models were sealed with epoxy resin. Figure 1a,c shows the detailed arrangement of the tunnel models with the reinforcements, moisture sensors, saturated glycerol electrodes, Ag/AgCl probes, and air inlet and outlet holes. The diameters of the air inlet and outlet holes were 10 mm. The air pump delivered air through the inlet hole into the ITT model’s cavity by the 10 mm diameter rubber tube, and then excluded the air from the outlet hole, thus realizing the air circulation in the model’s internal cavity. The tunnel models were subjected to standard curing (T = 20 ± 2 °C, RH ≥ 95%) for 21 d, the central 200 mm areas of the 4 sides of each model were retained as the test surface, and the remaining 2 sides and the edge areas of the 4 sides were sealed with 1 mm thickness epoxy resin, as shown in Figure 1b. Then, curing was continued up to 28 d.

Figure 2 shows the fabrication and testing processes of each ITT model, which includes the following: (1) model casting; (2) burying the rebars, saturated glycerol electrodes, and Ag/AgCl probes; (3) using wires to electrically connect the inner and outer rebar; (4) casting the top plate of the model; (5) installing the humidity sensors, as well as the inlet and outlet pipes; and (6) transferring the model into a pressure vessel to perform pressurized seawater erosion. The temperature of the room was controlled at 20 ± 2 °C during the test.

### 2.3. Pressurized Seawater Corrosion

An analysis of seawater samples obtained along the South China coast revealed a chloride ion concentration of 16.5 g/L; thus, the corrosion medium used in this study was based on sodium chloride at 10 times the concentration of simulated seawater, i.e., a chloride ion concentration of 165 g/L. To simulate the effect of the seawater pressure on chloride ion transport in the ITTs, the model was placed in a special corrosion vessel, and a pressurized device was used to maintain 6.5 atmospheres of pressure inside the vessel (comparable to the seawater pressure in the ITT of the Hong Kong–Zhuhai–Macao Bridge located at 40 m underwater) [36], as shown in Figure 2.

### 2.4. Electrochemical Test Methods

Electrochemical tests were performed at room temperature using a PARSTAT 2273 workstation to measure the self-corrosion potential, linear polarization curves, and AC impedance spectra. Pretest preparation: the ITT models to be tested were removed from the pressurized vessel, placed in a saturated calcium hydroxide solution (where the middle 200 mm test area on the side was just submerged in the solution), and left for 8 h before conducting the electrochemical tests, as well as electromagnetic shielding using a copper mesh cover during the electrochemical tests. The electrochemical test system was a three-electrode system with a saturated calomel electrode (SCE) as the reference electrode (RE), an external titanium mesh as the auxiliary electrode (AE), and a test rebar as the working electrode (WE). The actual corrosion area of a single test rebar in this experiment was calculated to be 50.24 cm^2^. The length, width, and thickness of AE were 150 mm, 150 mm, and 2 mm, respectively, and its exposed area was 705.56 cm^2^. The exposed area of the AE was about 14.0 times the area of the WE.

#### 2.4.1. Polarization Curves

According to the electrochemical weak polarization theory of metal corrosion, *R_p_* can be derived from the slope of the curve for the potential ΔE versus the current density Δ*i* at the corrosion potential (*E*−*E_corr_*) (shown in Figure 3):
(1)$$R_{p}={\left(\frac{∆E}{∆i}\right)}_{∆E\to 0}$$

The calculated *R_p_* can be used to determine the *i_corr_* using the Stern–Geary equation [37]:(2)$$i_{corr}=\frac{\beta_{a}\beta_{c}}{2.303(\beta_{a}{+\beta }_{c})}\frac{1}{R_{p}}=\frac{B}{R_{p}}$$
where Δ*E* is the applied polarization potential in mV; Δ*i* is the corresponding difference in the current density before and after polarization in μA/cm^2^; *R_p_* is the polarization resistance in Ω∙cm^2^; *i*_corr_ is the corrosion current density in μA/cm^2^; *B* is the Stern–Geary constant (which is taken as 52 mV and 26 mV when the test reinforcement in the concrete is in passivated and rust-activated states, respectively) [35]; and *β_a_* and *β_c_* are the Tafel constants for the anodic and cathodic polarization processes, respectively.

After the ITT models were eroded by pressurized seawater for 180 d and 365 d, linear polarization curves were generated by performing tests with a potential scan interval of *E_corr_* ±10 mV and a scan rate of 0.166 mV/s.

#### 2.4.2. AC Impedance Spectra

After the ITT models were eroded by pressurized seawater for 180 d and 365 d, AC impedance spectra (Nyquist plots) were measured by performing tests at *E_corr_* with the scan frequency interval set to 10 mHz–100 kHz and an applied AC voltage of 10 mV. The AC impedance spectra were then calculated by ZSimpWin V3.3 software, and the characteristics of the Nyquist plot with the best fit were used to select the equivalent electric circuit (EEC) shown in Figure 4 for fitting calculations, where the rebars were in passivation and corrosion states corresponding to *R_s_*(*C_c_*(*R_c_*(*Q_dl_*(*R_ct_*)))) and *R_s_*(*C_c_*(*R_c_*(*Q_dl_*(*R_ct_W*)))) [38,39], to determine the key electrochemical parameters associated with the corrosion of the rebars.

In Figure 4, *R_s_* denotes the solution resistance; in the high-frequency region, the time constants are *C_c_R_c_* (where *C_c_* denotes the capacitance of the concrete protective layer and *R_c_* denotes the resistance of the concrete protective layer) and *Q_dl_R_ct_*, considering the nonhomogeneity of both the concrete and the reinforcing steel surface (where the constant phase angle element CPE (denoted as *Q_dl_*) is used instead of the ideal double electric layer capacitance); *R_ct_* denotes the resistance to electron migration in the reinforcing steel to concrete interfacial region, which is affected by the passivation state of the reinforcing steel surface; and in the low-frequency region, *W* denotes the Warburg element introduced by diffusion control of corrosion of reinforcing steel.

## 3. Results

### 3.1. Self-Corrosion Potential E_corr_

Figure 5a shows the *E_corr_* distribution for the rebars in the outer and inner sides of the ITT models. At 180 d of pressurized seawater erosion, the mean *E_corr_* of the outer rebars in the ITT models was −434 mV with a variance of 350, and the mean *E_corr_* of the inner rebars was −294 mV with a variance of 148. At 180 d of erosion, the outer and inner rebars in the ITT models had stable *E_corr_* distributions and were in similar corrosion states. Compared to the inner rebars, the mean *E_corr_* of the outer rebars was negative 140 mV and 48% higher, indicating a clear tendency for the outer bars toward corrosion by seawater environment erosion. The relationship between *E_corr_* and the corrosion state of rebars in ASTM C876-2015 [40] was used to determine that at 180 d of erosion, the outer rebars had an *E_corr_* significantly lower than −276 mV and a probability of rebar corrosion of more than 90%, whereas the inner rebars had an *E_corr_* of approximately −276 mV, where the probability of rebar corrosion could not be determined. Thus, corrosion of the outer rebars in the ITT models was concluded to be significantly stronger than that of the inner rebars.

Figure 5b shows that at 365 d of erosion by pressurized seawater, the mean *E_corr_* of the outer rebars was −461 mV with a variance of 143, and the mean *E_corr_* of the inner rebars was −380 mV with a variance of 150. At 365 d of erosion, the outer and inner rebars had stable *E_corr_* distributions and similar corrosion states, where the mean *E_corr_* of the outer rebars was negative 81 mV and 21% higher compared with the inner rebars, indicating that the outer rebars were more severely corroded than the inner rebars by pressurized seawater erosion. The relationship between the *E_corr_* of the rebars and the corrosion state in ASTM C876-2015 [40] was used to determine that at 365 d of erosion, *E_corr_* of both the inner and outer rebars in the ITT models was significantly lower than 276 mV and the probability of rebar corrosion was more than 90%.

### 3.2. Linear Polarization Curves

Figure 6 shows the typical linear polarization curves of the outer and inner rebars in the ITT models at 180 d and 365 d of pressurized seawater erosion. The linear polarization curves for the outer and inner rebars were used to calculate the *i_corr_* of the rebars using Equations (1) and (2), and the calculation results were shown in Figure 7.

Figure 7a shows *i_corr_* of the outer and inner rebars in the ITT models at 180 d and 365 d of pressurized seawater erosion. At 180 d of erosion, the *i_corr_* of the outer rebars in the ITT models ranged from 0.365 to 0.562 μA/cm^2^ with a mean value of 0.455 μA/cm^2^ and a variance of 0.0038. The mean *i_corr_* of the outer rebars was close to 0.5 μA/cm^2^ [35], which was in a medium corrosion rate state, and the corrosion rate of the rebars was 0.0053 mm/year. The *i_corr_* of the inner rebars ranged from 0.218 to 0.290 μA/cm^2^ with a mean value of 0.260 μA/cm^2^ and a variance of 0.0004. The mean *i_corr_* value of the inner rebars was higher than the critical value of 0.1 μA/cm^2^ [35], and the rebars were in a low corrosion rate state with a corrosion rate of 0.0030 mm/year. The average corrosion rate of the outer rebars was 1.75 times higher than that of the inner rebars, i.e., the corrosion rate of the outer rebars was significantly higher than that of the inner rebars in the early process of corrosion in the ITT models.

Figure 7b shows *i_corr_* of the outer and inner rebars in the ITT models at 365 d of pressurized seawater erosion. At 365 d of erosion, the *i_corr_* of the outer bars in the ITT models ranged from 0.395 to 0.560 μA/cm^2^ with a mean value of 0.479 μA/cm^2^ and a variance of 0.0023, which was in a medium corrosion rate state, and the corrosion rate of the rebars was 0.0056 mm/year. The *i_corr_* of the inner bars ranged from 0.298 to 0.385 μA/cm^2^ with a mean value of 0.338 μA/cm^2^ and a variance of 0.0006, which was in a low corrosion rate state, and the corrosion rate of the rebars was 0.0039 mm/year. At 365 d of erosion, the average corrosion rate of the outer rebars in the ITT models was 1.43 times higher than that of the inner rebars, indicating that the corrosion rate of the outer rebars was still higher than that of the inner rebars. By comparing the corrosion rates of outer rebars and inner rebars in the ITT models, after erosion for 180 d and 365 d, the corrosion rate of outer rebars was always higher than that of inner rebars, and the corrosion rate of outer rebars remained at a higher rate as the corrosion time grew, while the corrosion rate of inner rebars gradually increased.

The analysis presented above shows that although both the outer and inner rebars were corroded at 180 d of erosion, the corrosion rate of the outer rebars was 1.75 times that of the inner rebars. This phenomenon occurred because the outer side of the ITT model was in direct contact with seawater, which obeys Fick’s second law of diffusion while penetrating into concrete, such that chloride ions continuously accumulated at the outer rebar surface and reached the critical content of 0.05% (mass percentage) [35] for the occurrence of corrosion earlier than the inner rebar. The corrosion of the rebars in concrete structures in a marine environment is an electrochemical process [11,12,13,41]: after the dense passivation film on the surface of the rebars is destroyed, the Fe substrate acts as the anode and loses electrons to generate Fe^2+^, whereby the substrate is dissolved and consumed; O_2_ and H_2_O in the solution on the surface of the reinforcing steel act as the cathode and gain electrons to generate OH^−^; at the same time, the Cl^−^ ions that have penetrated into the concrete from the eroding environment preferentially combine with Fe^2+^ to form soluble FeCl_2_, which accelerates the dissolution process of the iron matrix and increases the corrosion rate of the rebars.

During pressurized seawater erosion, the measured relative humidity (RH) of the air inside the cavity of the ITT models was maintained at approximately 97%, i.e., the seawater completely soaked through the tunnel wall, and the outer and inner rebar surfaces of the ITT models were in a water-saturated state. Considering the environmental conditions, the linear polarization curves show that the outer and inner rebars of the tunnel model were under anodic polarization control and that during the process of Cl^−^ transport from the outer to the inner side of the tunnel, the Cl^−^ content was higher on the surface of the outer rebars than on that of the inner rebars, de-passivation of the passivation film and rusting occurred for the outer rebars earlier than for the inner rebars, and the corrosion rate of the outer rebars was significantly higher than that of the inner rebars. At 180 d of erosion, both *E_corr_* and *i_corr_* of the inner rebars showed that the rebars were likely to have been de-passivated and had started to rust, but the surface Cl^−^ content was slightly higher than the critical value of 0.05% [35] and the corrosion rate of the rebars was low.

Through the analysis above, it can be seen that in ITTs, the reinforcing steel on the seawater side is weaker in durability than that on the cavity side. Therefore, in order to prolong the service life of ITTs, it is recommended to give priority to anti-corrosion methods for the reinforcing steel on the seawater side in the durability design of ITTs, such as applying polyurethane to the concrete surface on the seawater side to prevent seawater ingress [42], applying electrochemical rehabilitation to remove the sources of corrosion [43], and using cathodic protection [24,25] or surface coating epoxy [24,26] for the reinforcing steel on the seawater side. Given the large scale and high construction cost of ITT, the results of this work provide a theoretical basis for the adoption of a partial durability improvement method for the structure. Compared with the currently commonly used durability improvement method for the whole structure, the partial durability improvement method for the structure is more targeted and can reduce the construction and maintenance costs.

### 3.3. AC Impedance Spectra

Figure 8 shows the measured Nyquist plots of the ITT models after 180 d and 365 d of erosion in pressurized seawater. The Nyquist plots of each rebar exhibit an incomplete semicircular capacitance arc in both the high- and low-frequency regions and possibly a low-frequency tail. The diameters of the high- and low-frequency capacitance arcs correspond to the resistance of the concrete protective layer (*R_c_*) and the resistance to electron migration in the rebar–concrete interfacial region (*R_ct_*), and the low-frequency tail represents the material diffusion W for the corrosion reaction on the rebar surface. The Nyquist plots obtained at 180 d and 365 d of erosion were compared: the diameters of the high-frequency capacitive arcs for both the inner and outer rebars exhibited only a small increase with erosion age, indicating that *R_c_* increased slowly, which originated from the continuous hydration of the concrete protective layer and the slow increase in the resistance of the concrete protective layer; by comparison, the diameter of the low-frequency capacitive arc exhibited a significant decrease with erosion age, indicating *R_ct_* continuously decreased, the passivation film on the rebar surface was destroyed, and the corrosion rate of the rebars increased significantly.

To investigate the effect of the erosion age on the corrosion behavior of the inner and outer rebars of the ITT models, the EEC shown in Figure 4 was used to fit the Nyquist plots with an erosion age of 180 d, i.e., Figure 8a,c, and the results obtained from the fitting were listed in Table 5, according to which changes in the key parameters for electrochemical corrosion of the ITT models subjected to seawater erosion were analyzed.

Figure 9 shows the concrete resistance *R_c_*, resistance to electron migration *R_ct_*, and double-layer capacitance *Q_dl_* in the rebar−concrete interfacial region and the dispersion coefficient *n* obtained by fitting the Nyquist plots of the inner and outer rebars with the EEC. As shown in Figure 9a, at 180 d of erosion, the mean *R_c_* of the outer rebars in the ITT models was 15.2 kΩ∙cm^2^ with a variance of 9.896, while the mean *R_c_* of the inner rebars was 20.8 kΩ∙cm^2^ with a variance of 12.888. The *R_c_* of the outer rebars was 26.9% smaller than that of the inner rebars. The *R_c_* of the outer rebars was smaller than that of the inner rebars due to the difference in the test current path between the WE and AE, and the electrode test showed a “concrete protective layer” effect, i.e., the thickness of the concrete protective layer of the outer rebars was smaller than that of the inner rebars, and the contribution of the concrete to the polarization resistance was smaller for the outer rebars than for the inner rebars.

As shown in Figure 9a, at 180 d of erosion, the mean *R_c_* of the outer rebars in the ITT models was 15.2 kΩ∙cm^2^ with a variance of 9.896, while the mean *R_c_* of the inner rebars was 20.8 kΩ∙cm^2^ with a variance of 12.888. The *R_c_* of the outer rebars was 26.9% smaller than that of the inner rebars. The *R_c_* of the outer rebars was smaller than that of the inner rebars due to the difference in the test current path between the WE and AE, and the electrode test showed a “concrete protective layer” effect, i.e., the thickness of the concrete protective layer of the outer rebars was smaller than that of the inner rebars, and the contribution of the concrete to the polarization resistance was smaller for the outer rebars than for the inner rebars.

Figure 9b shows that the mean *R_ct_* of the outer rebars in the ITT models at 180 d of erosion was 39.9 kΩ∙cm^2^ with a variance of 21.012, and the mean *R_ct_* of the inner rebars was 83.5 kΩ∙cm^2^ with a variance of 41.144. The mean *R_ct_* of the outer rebars was only 48% of that of the inner rebars, corresponding to a significant difference between the two values. In a marine environment, the surface passivation film of the rebar in concrete is destroyed under erosion by seawater penetration; the iron matrix exposed to a pore solution with a high concentration of chloride ions is gradually dissolved; the resistance of harmful ions to migration to the surface of the iron matrix decreases; the resistance of the electrons lost by the iron matrix to reach the concrete through the ion interfacial zone via the rebars decreases; and the corrosion rate of the rebars increases [9]. In this study, the *R_ct_* of the outer rebars was found to be significantly lower than that of the inner rebars. For at 180 d of erosion, the chloride ion concentration on the surface of the outer rebars had exceeded the corrosion threshold of 0.05% [35], and the passivation film was damaged, resulting in the significant reduction of resistance to migration of electrons lost by the iron matrix to the concrete, and the rebars were corroded. At the same time, for the inner rebars, the passivation film was relatively intact, and no corrosion or only slight corrosion may have occurred.

Figure 9c shows that at 180 d of erosion, the mean *Q_dl_* of the outer rebars in the ITT models was 174/10^10^ Ω/cm^2^∙s^n^ with a variance of 81.684, and the mean *Q_dl_* of the inner rebars was 272/10^10^ Ω/cm^2^∙s^n^ with a variance of 177.340. Figure 9d shows that the mean *n* value of the outer rebars was 0.46 with a variance of 0.0026, which is lower than the critical value for steel corrosion of 0.6 [44], and the mean *n* value of the inner rebars was 0.80 with a variance of 0.0020, which is significantly higher than 0.6. During the erosion process, the passivation film of the rebars was destroyed and the surface unevenness increased significantly. The results for the electrochemical response process show that the *n* values decreased to approach and drop below the passivation critical reference value of 0.6 and the corrosion rate of the rebar continuously increased. The stable *n* value above 0.6 shows that a rebar is in a passivated state and corrosion is inhibited. The *n* values of the outer rebars in the ITT models were all below 0.6, indicating the occurrence of corrosion and a reduction in the surface flatness of all the bars, whereas all the *n* values of the inner rebars were above 0.6, indicating that the rebars may have remained in a passivated state.

Macias [45] found that *R_ct_* could be approximately equal to *R_p_* under certain conditions, but these two key parameters were different in most cases. Based on the data for the 32 sets of ITT models, the *R_ct_* data obtained from the AC impedance spectra were regressed against the *R_p_* data obtained from the linear polarization curves, and the results are shown in Figure 10.

The least squares method was used to obtain the following linear regression equation:(3)$$R_{p}=1.228R_{ct}$$

The coefficient of resolvability of the regression equation was 0.984, and the linear ratio of *R_ct_* to *R_p_* was 1.228, indicating a significant linear relationship. In addition, Figure 10 shows that *R_p_* was always greater than *R_ct_* and that the linear ratio between the two was in the range of 1.008 to 2.036, which is close to the results of Zhu’s study [46] on the electrochemical corrosion of reinforcing steel.

## 4. Conclusions

In this paper, the electrochemical corrosion behavior of reinforcing steel in the ITT models was investigated through indoor simulated pressurized seawater erosion tests, and the following main conclusions were obtained.

In the ITT models, the corrosion of the rebars on the seawater side happened earlier than the rebars on the cavity side, and the corrosion rate of the rebars on the seawater side was significantly higher than that of the rebars on the cavity side during the 365 d of erosion. The corrosion rate of the seawater side of the rebars maintained a high corrosion rate, while the corrosion rate of the cavity side of the rebars gradually increased with the growth of corrosion time.The electrochemical corrosion rates of the rebars on both the seawater side and the cavity side of the ITT models were controlled by anodic polarization, which indicated that chloride ion transport played a key role in the corrosion of the rebars, while oxygen transport had a small effect on the corrosion of the rebars.There was an obvious linear relationship between *R_p_* obtained by the linear polarization curves and *R_ct_* obtained by the AC impedance spectra in the reinforced concrete ITT models under the submarine environment, and the linear scale factor of 1.228 was obtained from the regression equation for both.Considering the reinforcing steel on the seawater side in the ITT is weaker in durability than that on the cavity side, it is recommended in the durability design to give priority to anti-corrosion methods for the reinforcing steel on the seawater side in order to prolong the service life of the ITT, such as applying polyurethane to the concrete surface on the seawater side to prevent seawater ingress, and using cathodic protection or surface epoxy for the reinforcing steel on the seawater side. Given the large size and high construction cost of the ITT, the results of this work show that it is feasible and more reasonable and economical to adopt durability enhancement methods for parts rather than for the entirety of the structure, while the corresponding service life assessment methods and life cycle cost analysis deserve further study.

## Figures and Tables

**Figure 1 materials-16-03300-f001:**
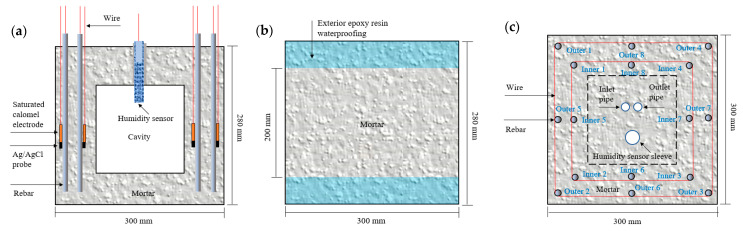
Sensor and rebar distribution (**a**), surface treatment (**b**), and rebar connection and air circulation holes (**c**) for the ITT models.

**Figure 2 materials-16-03300-f002:**
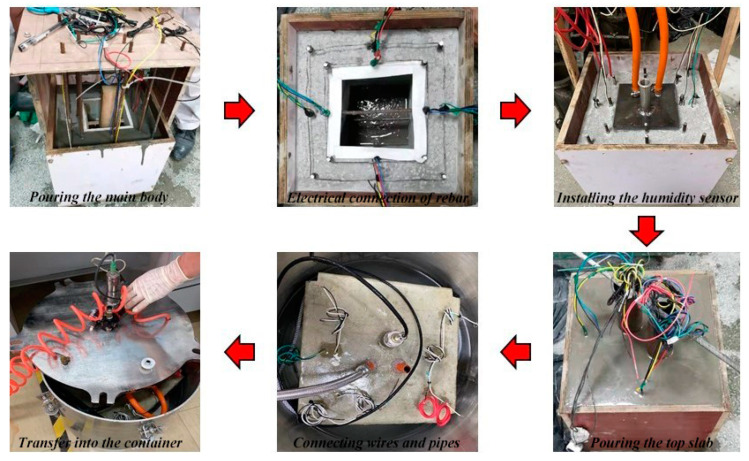
Fabrication and testing processes for the ITT models.

**Figure 3 materials-16-03300-f003:**
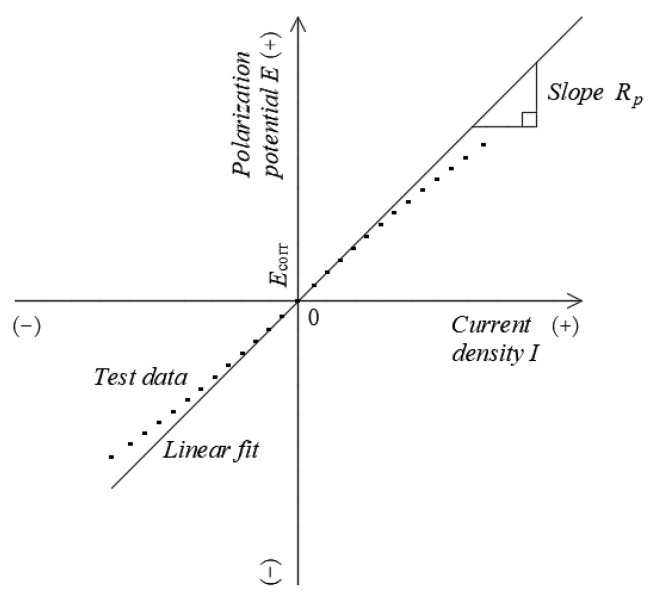
Calculating the *R*_p_ from the linear polarization curve [37].

**Figure 4 materials-16-03300-f004:**
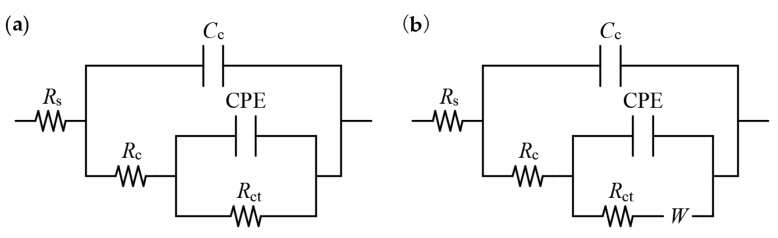
Rebar in a passivated state (**a**), rebar in a corroded state (**b**).

**Figure 5 materials-16-03300-f005:**
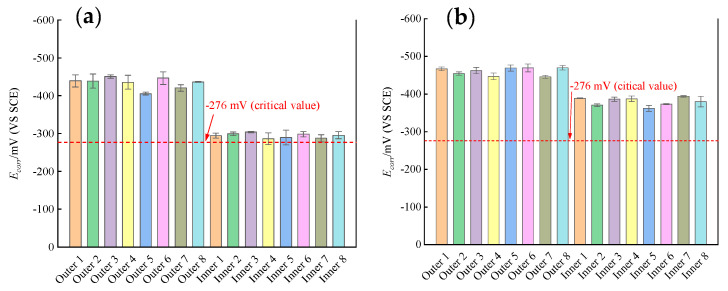
*E*_corr_ of the rebars in concrete after erosion 180 d (**a**) and 365 d (**b**).

**Figure 6 materials-16-03300-f006:**
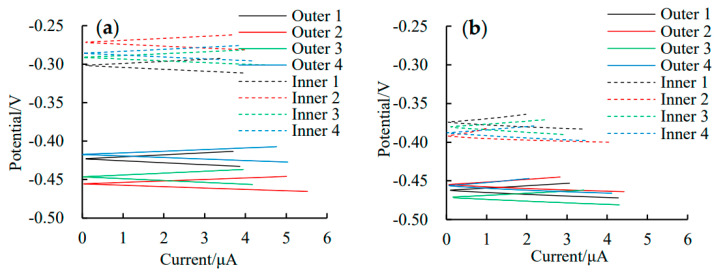
Linear polarization curves of the rebars in ITT models after erosion 180 d (**a**) and 365 d (**b**).

**Figure 7 materials-16-03300-f007:**
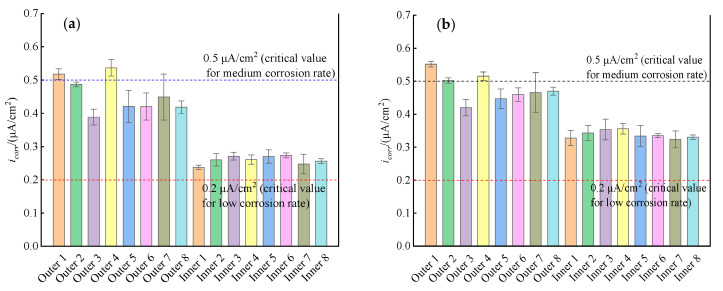
*i_corr_* of the rebars in ITT models after erosion 180 d (**a**) and 365 d (**b**).

**Figure 8 materials-16-03300-f008:**
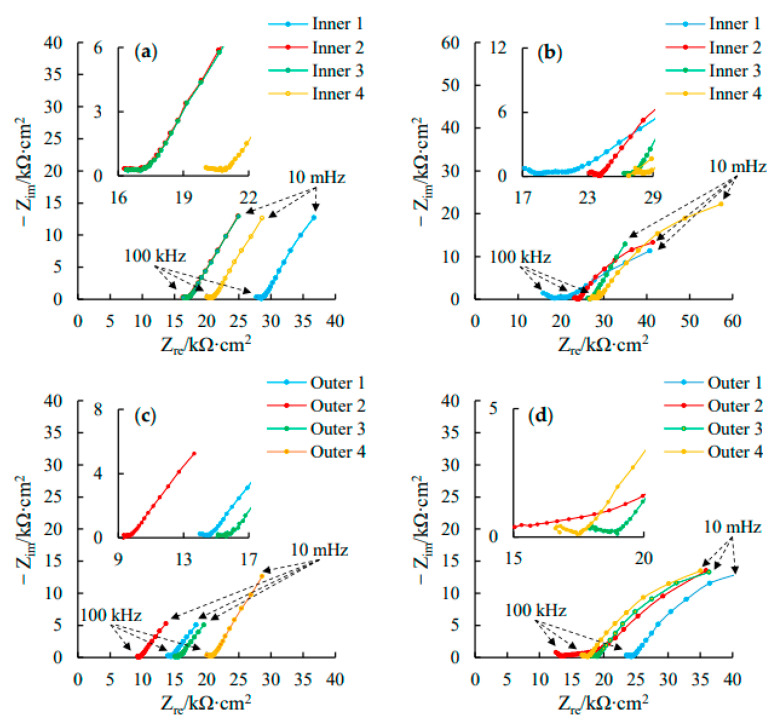
Nyquist diagrams of the inner rebars in ITT models after erosion 180 d (**a**) and 365 d (**b**); Nyquist diagrams of the outer rebars in ITT models after erosion 180 d (**c**) and 365 d (**d**).

**Figure 9 materials-16-03300-f009:**
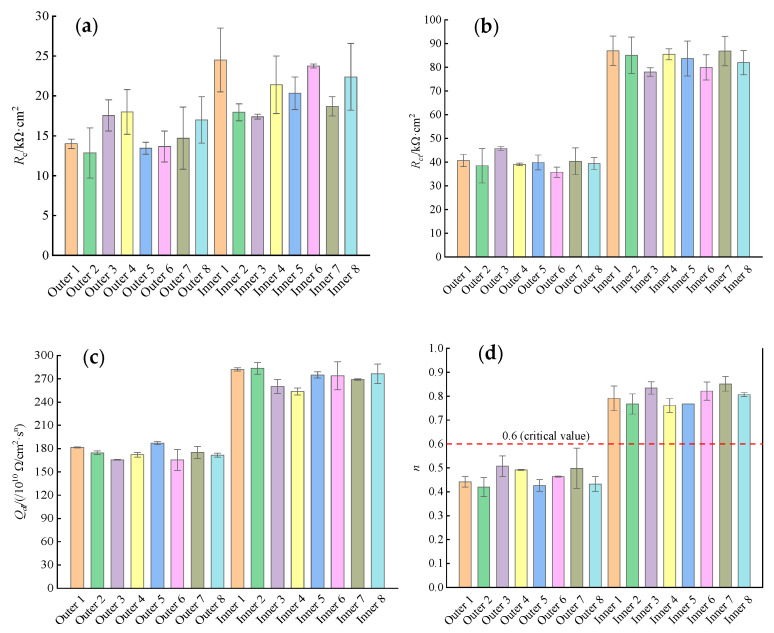
*R_c_* (**a**), *R_ct_* (**b**), *Q_dl_* (**c**) and *n* (**d**) of EEC fitting results for the Nyquist diagram of the rebars in ITT models.

**Figure 10 materials-16-03300-f010:**
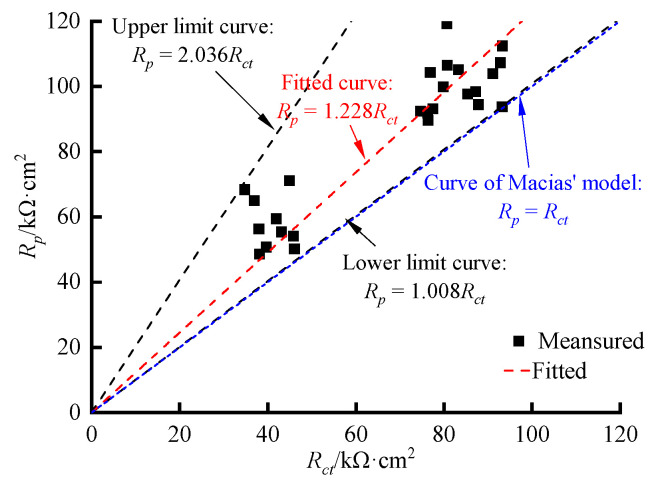
The relationship between *R_p_* and *R_ct_* of the rebars in ITT models.

**Table 1 materials-16-03300-t001:** Chemical composition of the cement (wt%).

LOI	SiO_2_	Al_2_O_3_	Fe_2_O_3_	CaO	MgO	SO_3_	K_2_O	Na_2_O	MnO	TiO_2_
1.37	19.26	4.70	3.28	64.91	3.46	2.05	0.83	0.14	-	-

**Table 2 materials-16-03300-t002:** Physical and mechanical properties of the cement.

Initial Setting Time/min	Final Setting Time/min	Specific Surface Area/(m^2^/kg)	Standard Consistency Water Consumption%	Stability	Compressive Strength/MPa		Flexural Strength/MPa	
					3 d	28 d	3 d	28 d
113	180	445.9	26.1	qualified	28.5	50.6	4.7	8.0

**Table 3 materials-16-03300-t003:** Mix ratio and properties of the mortar.

Materials/(kg/m^3^)			*w*/*c*	Fluidity /mm	28 d Flexural Strength/MPa	28 d Compressive Strength/MPa	28 d *D_RCM_* /(m^2^/s)
Cement	Sand	Water					
486	751	243	0.5	163	6.2	34.6	15.626/10^12^

**Table 4 materials-16-03300-t004:** Elemental composition of the rebar (wt%).

Fe	C	Mn	Si	P	S
98.197	0.168	1.425	0.160	0.023	0.027

**Table 5 materials-16-03300-t005:** Key parameters of the Nyquist plots fitted by EEC.

Number of Rebars	*R_c_*		*R_ct_*		*Q_dl_*		*n*	
	Fitting Result/ kΩ∙cm^2^	Standard Error	Fitting Result/ kΩ∙cm^2^	Standard Error	Fitting Result/(/10^10^ Ω/cm^2^∙s^n^)	Standard Error	Fitting Result	Standard Error
Model 1−Outer 1	14.6	4.3%	43.2	2.0%	181	10.9%	0.42	13.4%
Model 1−Outer 2	9.7	4.0%	31.2	2.2%	172	8.9%	0.38	10.3%
Model 1−Outer 3	15.6	3.8%	46.6	5.9%	166	21.0%	0.55	30.3%
Model 1−Outer 4	20.8	3.8%	38.5	5.6%	175	21.4%	0.49	30.9%
Model 1−Outer 5	14.2	4.6%	36.7	1.8%	189	5.8%	0.45	7.0%
Model 1−Outer 6	15.6	4.2%	33.6	1.7%	152	6.0%	0.46	9.0%
Model 1−Outer 7	18.6	8.4%	46.0	3.6%	167	17.3%	0.41	26.1%
Model 1−Outer 8	14.1	3.9%	36.9	8.7%	174	11.3%	0.46	3.2%
Model 1−Inner 1	28.5	11.4%	93.2	3.7%	284	20.1%	0.74	12.2%
Model 1−Inner 2	16.9	8.0%	77.4	1.7%	276	5.3%	0.81	6.4%
Model 1−Inner 3	17.1	8.2%	76.2	1.7%	269	11.4%	0.86	14.5%
Model 1−Inner 4	17.8	7.3%	83.2	4.1%	258	7.2%	0.79	8.8%
Model 1−Inner 5	22.4	7.8%	91.0	5.0%	279	5.1%	0.77	6.8%
Model 1−Inner 6	24.0	6.6%	85.3	3.7%	256	5.9%	0.86	9.2%
Model 1−Inner 7	17.5	6.1%	80.6	1.6%	268	5.1%	0.82	6.2%
Model 1−Inner 8	26.6	7.1%	87.1	1.8%	289	4.9%	0.80	5.1%
Model 2−Outer 1	13.4	15.9%	38.1	1.3%	182	13.4%	0.46	14.8%
Model 2−Outer 2	16.0	10.9%	45.7	11.3%	177	7.3%	0.46	5.6%
Model 2−Outer 3	19.5	7.0%	44.9	12.4%	165	5.6%	0.47	16.7%
Model 2−Outer 4	15.2	6.7%	39.6	11.5%	169	7.5%	0.49	3.7%
Model 2−Outer 5	12.7	3.8%	43.0	7.9%	185	14.0%	0.40	21.4%
Model 2−Outer 6	11.7	10.4%	37.9	14.9%	179	7.2%	0.47	8.1%
Model 2−Outer 7	10.8	4.8%	34.7	3.5%	183	11.4%	0.58	16.2%
Model 2−Outer 8	19.9	6.6%	41.9	9.5%	169	4.2%	0.40	9.7%
Model 2−Inner 1	20.5	4.0%	80.7	6.0%	280	17.9%	0.84	5.4%
Model 2−Inner 2	19.0	13.3%	92.7	13.6%	291	7.6%	0.73	15.6%
Model 2−Inner 3	17.7	6.1%	79.8	9.5%	251	7.2%	0.81	10.9%
Model 2−Inner 4	25.0	10.4%	87.8	9.8%	249	5.4%	0.73	9.2%
Model 2−Inner 5	18.3	9.4%	76.3	6.3%	271	4.6%	0.77	12.8%
Model 2−Inner 6	23.5	8.6%	74.6	6.7%	292	7.2%	0.78	6.2%
Model 2−Inner 7	19.9	8.4%	93.1	4.0%	270	14.0%	0.88	7.1%
Model 2−Inner 8	18.2	5.5%	76.8	10.3%	264	4.3%	0.82	9.7%

## Data Availability

The data used to support the findings of this study are available from the corresponding author upon request.

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
