# Peer review of "Corrosion Behavior of Reinforcing Steel in the Immersed Tube Tunnel (ITT) under Submarine Environment"

_materials, 2023, doi:10.3390/ma16093300_

Round 1

Reviewer 1 Report

The study is about corrosion testing of the far and near face reinforcement in ITT-based structures due to the interaction with seawater. The following are my comments towards improving the articles. 

A. Experimental method
1. The exact room temperature value should be indicated
2. What is the essence of adding mud into the mix as this material in missing in Table 3 as part of mortar composition?
3. What method was used to determine the consistency of the samples?

B. Literature review
4. Effects of temperature, cover and pH of pore solution of concrete on the corrosion process should be presented.

C. Discussion
5. The mechanism of passivation of the interaction of Cl ion with iron matrix (microstructure) should be presented separately and clearly. 

D. Conclusion
6. The specific import of this study on the durability design of ITT should be presented in the conclusion. 

The article could be understood in its present form and diction of English language employed.

Author Response

Dear Editors and reviewers:

Thank you for your comments concerning our manuscript. These comments are valuable and very helpful for revising and improving our paper and provide important guidance to our research. We have studied the comments carefully and have made corrections that we hope are met with approval. The revised portions are marked in red in the revised version. The main corrections in the paper and the responses to the comments please check the Word.

Reviewer 2 Report

1-   In this work the electrochemical corrosion behavior of reinforcing steel in the ITT models was investigated. The abstract section gives the summary of the work in a brief way which is appropriate.

2-   As we know the Cathodic protection is a technique used to control the corrosion of a metal surface by making it the cathodic side of an electrochemical cell. This technique has been investigated earlier and can be used in either submerged or buried structures. The originality and the scientific value of this article are acceptable however, I recommend authors to make clear what gap(s) did they intend to bridge with their research?

3-   The comparison between different techniques of corrosion control is also recommended (in addition to literature review details in Section 1).

4-   The methodology is very thorough however, there are differences between the mortar used in the present work and the concrete used in the actual projects (e.g. special concrete mix design utilized in Immersed Tube Tunnel (ITT) including special cement or admixture) as mentioned by authors. How you justify these differences?

5-   The authors also mentioned in the methodology that both mortar used in the present study and concrete for actual projects are close with respect to the transport behavior of aggressive agents (like concrete capillary water absorption action) and the corrosion behavior of reinforcing steel. Referring to some appropriate standards and strong justification is recommended.

6-   The presentation, analysis and clarity of results, data and figures are good. However, the authors are kindly recommended to provide some further details about the advantages of this technique. In addition, justification and comparison between the present study and other literature works should be written out in a much more detailed and comprehensive manner in Results Section.

7-   The conclusions perform the findings of the present study in a concrete manner and some protective measures for the reinforcing steel on the seawater side, in order to improve the seawater erosion resistance and extend the service life of ITT have been recommended. Some recommendations for future studies can be added at the end of this Section.

8-   The references are appropriate.

Author Response

(The authors gave the same response as above.)

Reviewer 3 Report

Authors of the paper investigated the corrosion behavior of reinforcing steel in the immersed tube tunel (ITT) under submarine environment. Lots of studies were performed on the steel behavior in a variety of environments. In the paper, the authors  conducted experiments on using electrochemical methods to investigate the corrosion behavior of reinforcing steel in the tunnels.

The main doubts are the „concrete structures buried deep in the sea that are fully saturated with seawater…”. What kinds are the real concrete structures, rectangular or circular in cross-section, (no view nor picture)… then, how deep are they burried, etc. The assumptions made by the authors do not refer to a real structure. They say, they scaled-down the ITT models subjected to erosion by pressurized seawater. How do you determine the > pressurized seawater<?

In Figure 1, what do you mean in (b) by >Surface treatment<? What surface, and what treatment? It seems to be a view, but… no information (?). Under (c) >Rebars connection…<, a 2D view would be more informative (?) 

What do you mean by >> the protective layer of the bars was set to 30 mm<<, was it formed by the cement//mortar? There are no information on the pipes used for the experiments… and their dimensions (?)

The key parameters indicated are Ecorr, Icorr, Rp, Rc, Rct and apparent diffusion coefficient (n). However, „I” usually stands for current (intensity), whereas „i” – for current density. The authors should be strict in this matter. In Table 1 title, there should be (wt%), not >>wt/%<<. Note MINUS sign throughout the paper, instead of your >>semi-minus<< used.

 For more details, see the PDF file with color highlights for your attention.

In general, English language may be accepted, but there are some faults and/or errors which should be improved - if the paper is to be considered for publication

For more details, see the PDF file with color highlights for the attention.

Author Response

Dear Editors and reviewers:

Thank you for your comments concerning our manuscript. These comments are valuable and very helpful for revising and improving our paper and provide important guidance to our research. We have studied the comments carefully and have made corrections that we hope are met with approval. The revised portions are marked in red in the revised version. The main corrections in the paper and the responses to the comments please see the attachment.
